# Metformin Restrains the Proliferation of CD4+ T Lymphocytes by Inducing Cell Cycle Arrest in Normo- and Hyperglycemic Conditions

**DOI:** 10.3390/biom14070846

**Published:** 2024-07-14

**Authors:** Ricardo Cartes-Velásquez, Agustín Vera, Bárbara Antilef, Sergio Sanhueza, Liliana Lamperti, Marcelo González-Ortiz, Estefanía Nova-Lamperti

**Affiliations:** 1School of Medicine, University of Concepcion, Concepcion 4070409, Chile; 2Molecular and Translational Immunology Laboratory, Department of Clinical Biochemistry and Immunology, Pharmacy Faculty, University of Concepcion, Concepcion 4070409, Chile; agvera2016@udec.cl (A.V.);; 3Laboratorio de Investigación Materno-Fetal (LIMaF), Departamento de Obstetricia y Ginecología, Universidad de Concepción, Concepción 4070409, Chile

**Keywords:** immunometabolism, CD4+ T cell, diabetes, metformin

## Abstract

CD4+ T lymphocytes play a key role in the modulation of the immune response by orchestrating both effector and regulatory functions. The effect of metformin on the immunometabolism of CD4+ T lymphocytes has been scarcely studied, and its impact under high glucose conditions, particularly concerning effector responses and glucose metabolism, remains unknown. This study aims to evaluate the effect of metformin on the modulation of the effector functions and glucose metabolism of CD4+ T lymphocytes under normo- and hyperglycemic conditions. CD4+ T lymphocytes, obtained from peripheral blood from healthy volunteers, were anti-CD3/CD28-activated and cultured for 4 days with three concentrations of metformin (0.1 mM, 1 mM, and 5 mM) under normoglycemic (5.5 mM) and hyperglycemic (25 mM) conditions. Effector functions such as proliferation, cell count, cell cycle analysis, activation markers and cytokine secretion were analyzed by flow cytometry. Glucose uptake was determined using the 2-NBDG assay, and levels of glucose, lactate, and phosphofructokinase (PFK) activity were assessed by colorimetric assays. Metformin at 5 mM restrained the cell counts and proliferation of CD4+ T lymphocytes by arresting the cell cycle in the S/G2 phase at the beginning of the cell culture, without affecting cell activation, cytokine production, and glucose metabolism. In fact, CD69 expression and IL4 secretion by CD4+ T lymphocytes was higher in the presence of 5 mM than the untreated cells in both glucose conditions. Overall, metformin inhibited proliferation through mechanisms associated with cell cycle arrest, leading to an increase in the S/G2 phases at the expense of G1 in activated CD4+ T lymphocytes in normo- and hyperglycemic conditions. Despite the cell cycle arrest, activated CD4+ T lymphocytes remained metabolically, functionally, and phenotypically activated.

## 1. Introduction

The intricate relationship between the immune response and metabolism strongly influences physiological and pathological conditions across organs, cells, and molecules. This recognition has propelled immunometabolism into cutting-edge translational research, advancing our comprehension of how cellular metabolism affect the immune responses and therefore the development of novel therapies for metabolic and immune disorders [1]. Immunometabolism broadly encompasses two main facets: the impact of immune cells on the body’s metabolism, and the intracellular metabolic pathway alterations in immune cells upon activation [1,2,3]. The six most scrutinized metabolic pathways in immunometabolism include glycolysis, oxidative phosphorylation, the pentose phosphate pathway, fatty acid synthesis, fatty acid oxidation, and amino acid metabolism [2]. Immunometabolic changes at the cellular level hinge on the cell type, stimulus, environmental conditions, and interactions with other cell types. These changes significantly influence how immune cells obtain energy, and the metabolites from these pathways play a crucial role in determining immune cell functionality—a pivotal aspect in understanding immune responses [1,2,3].

The notion that resting immune cells opt for more efficient pathways for energy production, such as oxidative phosphorylation, is generally accepted. Upon activation, these cells swiftly switch to aerobic glycolysis, a less efficient but faster energy-producing process that generates various metabolites essential for immune cell proliferation. Memory or regulatory T cells, however, revert to relying mainly on oxidative phosphorylation. These metabolic shifts are subject to modification, opening the door to the metabolic and functional reprogramming of immune cells [1]. CD4+ T lymphocytes play a pivotal role in adaptive immune responses, contributing to both effector and regulatory components. Diverse CD4+ T cell subpopulations with specific functions and different immunometabolism collaborate synergistically for an effective adaptive immune response [4,5]. For instance, regulatory T cells, during activation and differentiation, exhibit lower utilization of the glycolytic pathway (compared to other T cells) in favor of increased oxidative phosphorylation and free fatty acid oxidation, whereas the primary metabolic pathway during effector CD4+ T cell activation is the glycolytic pathway [5].

As one of the most relevant targets in immunometabolism, CD4+ T lymphocytes [6,7,8,9,10] play a central role in both effector and regulatory functions. In recent years, metformin effects beyond its hypoglycemic action have been described, with potential uses in cardiovascular and neurological contexts [11]. Metformin has gained attention for its therapeutic potential in autoimmune diseases [2,12,13,14], as an adjuvant in cancer treatment [15,16] and HIV management [17], and even for COVID-19 [18,19,20]. CD4+ T lymphocytes in human and mouse thyroiditis models have shown increased activation of the mTOR/HIF-1α/HK2/glycolysis pathway, which was reduced by metformin. Furthermore, the Th17/Th1 ratio decreased with the metformin agent, implying not only metabolic but also immunophenotypic changes [21]. In a systemic lupus erythematosus model, metformin was shown to inhibit pSTAT1 phosphorylation and its binding to IFN-g-responsive elements independently of AMPK and mTOR [22].

Regardless of the pathways, the evidence consistently indicates that metformin has a modulatory effect on the effector differentiation of CD4+ T cells, inhibiting Th17 and Th1 responses. This appears to be linked to a reduction in glycolytic pathway activity. Despite the growing evidence supporting the potential benefits of metformin on immunity, addressing the knowledge gap regarding its effect in humans under conditions of high glucose typical in individuals using metformin for diabetes mellitus treatment is crucial. Most studies have focused on mouse models and employed glucose concentrations in the range of 100 to 200 mg/dL, which does not reflect the actual clinical situation of these patients. This study aims to evaluate the effect of metformin on the glycolytic pathway and immune activation of activated CD4+ T lymphocytes in normal and high glucose (450 mg/dL).

## 2. Materials and Methods

### 2.1. Isolation of CD4+ T Lymphocytes

CD4+ T cells were isolated from six healthy donors. Individuals with any disease, metabolic, endocrine, hematological, or immunological disorder, and those with obesity or malnutrition were excluded. Individuals who had used medication affecting glycemia in the last 6 months or who had a history of difficulty in obtaining blood samples via venous puncture were also excluded. Donors gave their consent in accordance with the University of Concepcion Ethics Committee, reference number project PAI79170073, and blood sample collection was scheduled for the same day, around noon, and collected through venous puncture. Mononuclear cells were isolated by density gradient centrifugation (400× *g*, 20 min) at room temperature with Lymphoprep (Axis Shield, Dundee, Scotland). T lymphocytes were enriched using the human CD3+ T Memory Cell Isolation Kit (Miltenyi Biotec, Bergisch Gladbach, Germany), and sorted after anti-CD4 staining-FITC (BioLegend, CA, USA) with BD FACSAria III sorter (BD Biosciences, CA, USA).

### 2.2. Culture Conditions

For cell activation, RPMI 1640 serum (Gibco, MA, USA) supplemented with 10% fetal bovine serum, 500 IU/mL of IL-2 (ProLeukin, CA, USA), and anti-CD3/CD28 beads (1:4 ratio) (Life Technologies, CA, USA) with metformin (0 mM, 0.1 mM, 1 mM, or 4 mM) and glucose in the medium (5.5 mM or 25 mM) were used at 37 °C, 5% CO_2_ for 96 h.

### 2.3. Glucose Uptake

Glucose uptake was assessed using the fluorescent marker 2-(N-(7-nitrobenz-2-oxa-1,3-diazol-4-yl)amino)-2-deoxyglucose (2-NBDG, 10 µM) (ThermoFisher Scientific, CA, USA), according to the manufacturer’s instructions. Briefly, cells were washed (300× *g*, 10 min) and incubated in PBS for 30 min at 37 °C, 5% CO_2_ in the dark. Then, 2-NBDG (10 µM) was added, and cells were incubated for 45 min, 37 °C, 5% CO_2_ in the dark. Samples were acquired using the LSR Fortessa system (BD Biosciences, CA, USA), and analyzed with FlowJo software version 8 (FlowJo LLC, OR, USA).

### 2.4. Lactate Production

Lactate production was determined using the Lactate Kit (COD 12736; Biosystems, Barcelona, Spain), according to the manufacturer’s instructions. Briefly, a standard curve was prepared with the human calibrator AX125 (Biosystems, Barcelona, Spain). Measurements were then performed using a Synergy 2 plate reader (BioTek Instruments, Bad Friedrichshall, Germany) at a wavelength of 600 nm, and data were analyzed with Gen5 software version 2.0 (Agilent, CA, USA). Concentrations of lactate were normalized to the number of cells in each measurement using CountBright™ beads (Life Technologies, CA, USA).

### 2.5. Cell Proliferation Assay

Cell proliferation was assessed using the CellTrace™ Violet Kit (Life Technologies, CA, USA) following the manufacturer’s instructions before activation and culture conditions. Briefly, CD4+ T lymphocytes were washed with PBS (300× *g*, 10 min) and stained with CellTrace Violet (20 µM) for 20 min at 37 °C in an atmosphere with 5% CO_2_. The reaction was stopped by adding 2 mL of cold RPMI medium supplemented with 10% FBS. Samples were acquired using the LSR Fortessa system (BD Biosciences, CA, USA). For proliferation analysis, the FlowJo proliferation platform (FlowJo LLC, OR, USA) was used, allowing the calculation of the number of undivided cells and four proliferation indices: division index (total number of divisions/number of cells at the start of culture), proliferation index (total number of divisions/cells that entered division), expansion index (total number of cells/cells at the start of culture), and replication index (total number of divided cells/cells that entered division).

### 2.6. Activation Marker Staining

For activation markers, cells were labeled with the following antibody panel: anti-CD25 (PE-Cy7), anti-CD69 (FITC), anti-CD109 (PE), and anti-CD4 (APC-Cy7) (All BioLegend, CA, USA), and IR-LIVE/DEAD (ThermoFisher Scientific, CA, USA) for 20 min at 4 °C in the dark. Samples were acquired using the LSR Fortessa system (BD Biosciences, CA, USA), and analyzed with FlowJo software (FlowJo LLC, OR, USA).

### 2.7. Cell Cycle Analysis

Propidium iodide staining was used to determine the phases of the cell cycle. Cells were washed with PBS (300× *g*, 10 min), and 500 µL of 70% ethanol was added while vortexing at maximum intensity to prevent cell clumping. Samples were kept in the dark at 4 °C for 30 min and then washed with PBS (300× *g*, 10 min) twice. Subsequently, a solution of 50 µL of propidium iodide (Life Technologies, USA) in 1700 µL of buffer (Life Technologies, USA) was prepared. Cells were resuspended in 100 µL of this solution for analysis by flow cytometry. Samples were processed using the LSR Fortessa system (BD Biosciences, USA), and resulting files were analyzed with FlowJo software (FlowJo LLC, USA).

### 2.8. Cytokine Secretion

Cytokines IL-17, IFN-g, TNF-a, and IL-4 were measured with the BD Cytometric Bead Array (CBA) Human Chemokine Kit (BD Biosciences, USA). Samples were processed using the LSR Fortessa system (BD Biosciences, USA), and resulting files were analyzed with FCAP Array software version 3.0 (BD Biosciences, CA, USA) to determine the concentration of each analyte, expressed in pg/mL.

### 2.9. Statistical Analysis

Statistical analysis was performed using Prism 10 software (GraphPad Software, MA, USA). Descriptive statistics were generated for each described condition and presented through scatter plots including individual points and error bars representing the mean and standard error. All data were grouped according to glucose concentrations (5.5 mM and 25 mM). For inferential statistical analysis, a two-way repeated measures ANOVA was employed. Multiple comparisons were made between metformin concentrations (0 mM, 0.1 mM, 1 mM, and 5 mM) for glucose groups (5.5 mM and 25 mM) separately using Fisher’s LSD (Least Significant Difference) test. Additionally, comparisons were made between glucose groups for each of the metformin concentrations. In all cases, statistical significance was considered when *p* < 0.05. The *p*-values are presented as follows to indicate the level of significance: * (*p* < 0.05), ** (*p* < 0.01), *** (*p* < 0.001), and **** (*p* < 0.0001).

## 3. Results

### 3.1. Metformin Affects Cell Counts and Proliferation of In Vitro Activated CD4+ T Lymphocytes under Normal and High Glucose

Metformin has been widely used in patients with diabetes, prediabetes, and other diseases to regulate glucose metabolism by different mechanisms. Since this metabolic pathway is key in the effector response of CD4+ T lymphocytes, we studied the in vitro effect of metformin upon cell proliferation in CD4+ T lymphocytes. We first analyzed cell counts after 4 days of T cell activation under normal and high glucose using three concentrations of metformin. We observed that the presence of 5 mM metformin significantly reduced cell counts under normal and high glucose concentrations in comparison to other conditions (Figure 1A), demonstrating that a high concentration of metformin reduced cell numbers during the 4 days of activation. To evaluate whether metformin affected cell proliferation, the same conditions were analyzed using a cell viability dye with a cell proliferation probe. The data showed that under 5.5 mM and 25 mM of glucose, the percentage of non-divided cells at 5 mM metformin was significantly augmented compared to non-treated cells or cells treated with lower concentrations of metformin (Figure 1B). Then, using the FlowJo proliferation platform, we observed that under both glucose conditions, 5 mM metformin significantly reduced the division index (Figure 1C), proliferation index (Figure 1D), replication index (Figure 1E), expansion index (Figure 1F), and number of cell generations (Figure 1G) in a concentration-dependent manner. Overall, our data demonstrated that the presence of 5 mM metformin in vitro inhibited the proliferation of 4-day-activated CD4+ T lymphocytes under normal and high glucose.

### 3.2. Metformin Arrests the Cell Cycle of In Vitro-Activated CD4+ T Lymphocytes under Normal and High Glucose

We observed that activated CD4+ T lymphocytes in the presence of a high metformin concentration exhibited a proliferation impairment, resulting in reduced cell counts; thus, we evaluated whether this effect was associated with the capacity of metformin to affect cell cycles, as previously shown in cancer cells [23,24]. Using propidium iodide, a fluorescent intercalating DNA dye, we observed the percentage of CD4+ T lymphocytes in diverse phases of the cell cycle interface (Figure 2A). Under both glucose concentrations, 5 mM metformin induced a significant decrease in the percentage in the G1 phase (Figure 2B), with a significant increment in the percentage in the S (Figure 2C) and G2 phases (Figure 2D) in comparison to non-treated cells and cells treated with lower concentrations of metformin. These data demonstrated that the effect of metformin upon the cell proliferation of activated CD4+ T lymphocytes under normal and high glucose was due to cell cycle arrest.

### 3.3. Despite Affecting Cell Cycle, Metformin Up-Regulates the Activation Marker CD69 in CD4+ T Lymphocytes In Vitro Activated under Normal and High Glucose

After analyzing the effect of metformin upon cell count, proliferation, and cell cycle in activated CD4+ T cells, we evaluated the impact of metformin on cell activation by analyzing CD69 and CD25 expression. We observed that under both glucose concentrations, 5 mM metformin significantly increased CD69 expression in comparison to other concentrations (Figure 3A,B). Then, the expression of CD25, corresponding to the alpha chain of the IL-2 receptor, pivotal in the lymphocyte proliferative process, was evaluated. This marker was highly expressed in activated lymphocytes under both glucose conditions; however, 5 mM metformin induced a significant reduction compared to other concentrations (Figure 3C,D). Discrepancies in CD69 and CD25 expression reveal opposite effects of metformin on distinct activation markers, and thus the analysis of other effector functions is required to understand whether metformin affects T cell responses.

### 3.4. Metformin Modulates Cytokine Secretion in Activated CD4+ T Lymphocytes

Following our exploration of the impact of metformin on cell proliferation and activation markers, we analyzed the modulatory capacity of metformin on the secretion of Th1, Th2, and Th17 cytokines in activated CD4+ lymphocytes. For Th1 cytokines, under high glucose, 5 mM metformin reduced the secretion of IFN-g in comparison with the untreated (Figure 4A); however, no differences were observed for TNF-a secretion (Figure 4B). For Th2 cytokines, we observed that under both glucose concentrations, 5 mM metformin induced a significant increase of IL-4 compared to other concentrations (Figure 4C). For Th17 cytokines, under 5.5 mM glucose, both 0.1 mM and 5 mM metformin concentrations induced a significant decrease of IL-17 (*p* < 0.05) compared to the non-treated condition (Figure 4D); however, at 25 mM glucose, no significant differences were observed (Figure 4D). Overall, the data suggest that metformin promotes Th2 responses and inhibits Th1 and Th17 responses, even when the cell cycle is arrested, indicating that the cells are metabolically and functionally active.

### 3.5. Metformin Promotes Glucose Uptake and Lactate Production in Activated CD4+ T Lymphocytes

Our previous results strongly suggested that CD4+ T lymphocytes were functionally active, despite cell cycle arrest, and thus we analyzed the effect of metformin on glucose metabolism in the T cells. First, glucose uptake by activated CD4+ T lymphocytes was assessed by measuring 2-NBDG uptake after a 4-day culture under normal- and high-glucose concentrations (Figure 5A,B). At 5.5 mM glucose condition, only the 5 mM metformin concentration exhibited a significant increase in glucose uptake compared to the metformin-free control. At 25 mM glucose conditions, the 5 mM metformin concentration induced a significant uptake compared to other metformin concentrations and the untreated sample. For lactate (Figure 5C), in both glucose conditions, the 5 mM metformin concentration significantly increased lactate production relative to other concentrations and the control sample. These data suggest that metformin promotes glucose metabolism despite cycle arrest.

## 4. Discussion

In this study, we evaluated the effect of metformin on the modulation of effector functions and glucose metabolism of CD4+ T lymphocytes under normal- and high-glucose conditions. Our data showed that metformin at 5 mM restrained the cell counts and proliferation of CD4+ T lymphocytes, by arresting the cell cycle in S/G2 phase at the beginning of the cell culture, without affecting cell activation, cytokine production, and glucose metabolism. In fact, CD69 expression and IL4 secretion by CD4+ T lymphocytes was higher in the presence of 5 mM of the drug than in the untreated cells, in both glucose conditions. We also observed that metformin reduced Th1 and Th17 cytokines at high concentrations, despite not reducing glucose uptake and lactate production. Overall, metformin inhibited proliferation through mechanisms associated with cell cycle arrest, leading to an increase in the S/G2 phases at the expense of G1 in activated CD4+ T lymphocytes in normo- and hyperglycemic conditions. Despite the cell cycle arrest, activated CD4+ T lymphocytes remained metabolically, functionally, and phenotypically activated.

Regarding the proliferative activity in our experiments, our findings consistently demonstrate that metformin inhibits CD4+ T lymphocyte proliferation. This inhibitory effect has previously been observed in a mouse model of insulitis [25], where the response was dose-dependent within the 0–10 mM range of metformin. The results of this study add evidence by showing that this inhibitory effect on proliferation also occurs in human CD4+ T lymphocytes. The inhibition of T lymphocyte proliferation could be beneficial in certain contexts, such as in autoimmune diseases where the suppression of an excessive immune response is desired [9,25,26,27,28,29,30,31,32]. However, it is important to note that T lymphocyte proliferation is essential for an effective immune response in other contexts, such as in defense against infections, particularly in sepsis conditions [33]. Therefore, the effects of metformin on T lymphocyte proliferation must be carefully weighed based on the clinical context and patient needs.

Cell cycle arrest was observed in CD4+ T lymphocytes at 5 mM of metformin. It has been reported that metformin inhibits the proliferation of myeloma cell lines in RPMI8226 and U266 cells by inducing G0/G1 cell-cycle arrest at 5 mM and 20 mM for 24 h [23]. In SKM-1 cells, a myelodysplastic syndrome to the acute myeloid leukemia cell line, metformin inhibited cell proliferation instead of promoting apoptosis, by arresting cell cycle at G0/G1 when metformin was used at 5 mM and 20 mM, in comparison with the untreated group [24]. Another study showed, in osteosarcoma 143B and U2OS cell lines, that metformin arrested cells in G2/M, reducing the percentages of cells in G0/G1, promoting apoptosis at 10 mM and 20 mM, but not at 5 mM [34]. In four breast cancer cell lines, MCF-7, MCF-7/713, BT-474, and SKBR-3 cells, metformin induced cell cycle arrest at 2 mM, 10 mM, and 50 mM [35]. We found that arrest would occur in the S/G2 phases, with no differences between high and normoglycemia conditions. It is relevant to note here that activated CD4+ T lymphocytes and cancer cells share the quality of being highly proliferative cells. Similar to breast cancer [36] or bladder cancer [37], this similarity suggests that the anti-proliferative effect of metformin might be related to its ability to influence the regulation of cyclins. The regulation of the cell cycle is a highly complex and tightly regulated process involving a series of proteins, including cyclins and cyclin-dependent kinases [38]. If metformin selectively affects the expression or activity of these proteins, it could explain the cell cycle arrest observed in human CD4+ T lymphocytes.

Metformin notably increases the percentages of CD69+ cells but reduces CD25 levels. However, these results have previously been described in a mouse model of influenza, where treating isolated CD4+ T cells with metformin 2 mM resulted in the inhibition of the oxygen consumption rate (OCR) and the modulation of the CD69 and CD25 [39]. In that study, the decrease in CD25 was interpreted as an indicator of anti-proliferative effects, while the effect of CD69 was associated with an increase in resident memory T lymphocytes. Recent evidence has shown that resident memory T lymphocytes have better capacity at adjusting the metabolism to an inflammatory environment. This facilitates their ability to develop a faster and more effective response once activated, compared to circulating T lymphocytes [40]. It is important to note that, while these results are consistent with previous findings and suggest a coherent interpretation of the effects of metformin on CD4+ T lymphocytes, further research is required to fully understand the underlying mechanisms and their clinical relevance in different immunological contexts. Regarding cytokine production, we found results that are consistent with previous evidence showing that metformin stimulates Th2 responses and inhibits Th17 and Th1 responses [25,30,33]. In their study, using a mouse model of insulitis in non-obese diabetic (NOD) mice, Duan et al. [25] discovered that administering metformin to NOD mice notably alleviated autoimmune insulitis. Additionally, the treatment led to a notable reduction in the number of pro-inflammatory IFN-g+ and IL17+ CD4 T cells in the spleens of NOD mice. Conversely, there was a significant increase in the percentage of regulatory IL-10+ and Foxp3+ CD4 T cells observed. Park et al. [30] found that administering metformin intraperitoneally, activating AMPK signaling significantly, improved the severity of acute graft-versus-host disease and reduced mortality. This was accompanied by decreased levels of Th1 and Th17 cells, increased levels of Th2 and Treg cells, and the enhanced conversion of Th17 cells into Treg cells through autophagy. Metformin’s mortality-reducing effect was associated with the inhibition of the mammalian target of rapamycin/signal transducer and the activator of transcription 3 pathway. Similarly, Zhao et al. [21] observed an elevated expression of HK2 and PKM2 in cultured mice with subacute thyroiditis (SAT) CD4+ T cells compared to the control group. A significant activation of the mTOR and HIF-1α pathways was noted in SAT mice, with a reduced expression of HIF-1α in the group treated with 2DG. Treatment with 2DG and/or metformin led to a significant decrease in the ratio of Th17 and Th1 T cells.

Despite cell cycle arrest, the glucose metabolism of activated CD4+ T lymphocytes was not impaired. In fact, higher glucose uptake and lactate production were observed in the presence of metformin. Our findings agree with the results previously reported by Yin et al. [41], who found elevated glycolysis and mitochondrial oxidative metabolism in CD4(+) T cells from lupus-prone B6.Sle1.Sle2.Sle3 (TC) mice compared to non-autoimmune controls. In vitro, both the mitochondrial metabolism inhibitor metformin and the glucose metabolism inhibitor 2-deoxy-D-glucose (2DG) reduced IFN-g production, albeit at different activation stages. Treating TC mice and other lupus models with a combination of metformin and 2DG normalized T cell metabolism and reversed disease biomarkers. Moreover, CD4(+) T cells from SLE patients displayed enhanced glycolysis and mitochondrial metabolism correlating with their activation status, and metformin significantly reduced their excessive IFN-g production in vitro. However, another study in a subacute thyroiditis mouse model found that metformin and 2-DG reduced the glycolytic activity [21].

After activation, glucose uptake by T lymphocytes is regulated by the translocation of the GLUT1 transporter [42,43]. However, it has been explained that metformin stimulates GLUT4 transporter translocation [44]. This could be associated with the increment in glucose uptake and lactate production. Regarding the secretion of glycolysis-derived products, we observed that 5 mM metformin produced the greatest increase in lactate levels, and that these levels were higher in high-glucose conditions. This result is consistent with the acidification rate described in the literature [10,41]. Tan et al. [10] observed that compared to either metformin or 2-DG treatment alone, dual treatment with metformin + 2-DG resulted in an enhanced suppression of GlycoPER in human CD4+ T cells activated for 24 h with anti-CD3/anti-CD28. While metformin treatment alone reduced the OCR as expected, this effect was not further potentiated by metformin + 2-DG treatment. Meanwhile, Yin et al. [41] observed that in a lupus mice model, metformin dose-dependently (0 to 1 mM) decreased the oxygen consumption rate and increased the extracellular acidification rate at a high concentration, likely serving as an alternative pathway to generate ATP. Lactate production is a direct consequence of glycolysis, and an increase in lactate production is considered an indication of higher glycolytic activity. Therefore, these findings support the idea that metformin stimulates the glycolytic pathway in human CD4+ T lymphocytes. However, it is relevant to note that these results also raise the possibility that metformin may have the ability to stimulate lactate dehydrogenase activity at the expense of pyruvate kinase, which could influence lactate production [45]. This specific aspect may require further investigation to fully understand the mechanisms underlying the effects of metformin on lactate production in human CD4+ T lymphocytes.

Several limitations of this study need to be acknowledged. Firstly, in line with most of the literature [10,21,25,30], it is important to note that significant effects occurred with metformin at concentrations of 5 mM, which significantly exceeded the serum concentrations in individuals undergoing treatment with this drug, typically around 5 μM [46]. However, given that the culture conditions are different from in vivo conditions, it is challenging to confirm or rule out whether the effects of these 5 mM concentrations in the culture corresponded to those in vivo at 5 μM. Nonetheless, this could also be considered a strength, as our results can be compared with the literature, which typically considers metformin concentrations ranging from 1 to 10 mM.

Secondly, CD4+ T lymphocytes in their various subpopulations have diverse roles and interactions with other immune cells, so this study only shows the direct effects of metformin on CD4+ T lymphocytes, but does not cover effects that may be mediated by other populations (e.g., CD8+). Additionally, the effects on cytokines may be attributed to the increase in Treg and the reduction in Th17 subpopulations. While this study did not directly assess these aspects, they have previously been described in the literature for hepatic tissue [47], airway inflammation [27], and bone metabolism [48].

Thirdly, we only included two glucose concentrations representing a physiological trend (5.5 mM or 100 mg/dL) versus a pathological condition (25 mM or 450 mg/dL) that would be expected in a diabetic patient with poor glycemic control. However, by not evaluating intermediate concentrations, we cannot assert linearity or dose–response in the differences found between these concentrations.

Despite these limitations, these results contribute evidence to the field of immunometabolism using human CD4+ T lymphocytes evaluated under high glucose conditions, which had not been directly assessed in previous studies. The variations observed in the response to metformin based on glucose concentration may have significant clinical implications. For instance, in patients with type 2 diabetes, glucose levels can vary widely, and therefore the effects of metformin could differ from one individual to another. However, it is essential to consider that the experimental conditions evaluated in this study involved a short exposure period (4 days) and lacked any immunological challenges such as infection or cancer. This aspect is crucial for a more accurate extrapolation of these results to clinical practice, and in demonstrating the need for personalized medical care and the importance of considering the patient’s glucose levels when making therapeutic decisions related to metformin.

## 5. Conclusions

Metformin inhibited proliferation through mechanisms associated with cell cycle arrest, leading to an increase in the S/G2 phases at the expense of G1 in activated CD4+ T lymphocytes in normo- and hyperglycemic conditions. Despite the cell cycle arrest, activated CD4+ T lymphocytes remain metabolically, functionally, and phenotypically activated.

## Figures and Tables

**Figure 1 biomolecules-14-00846-f001:**
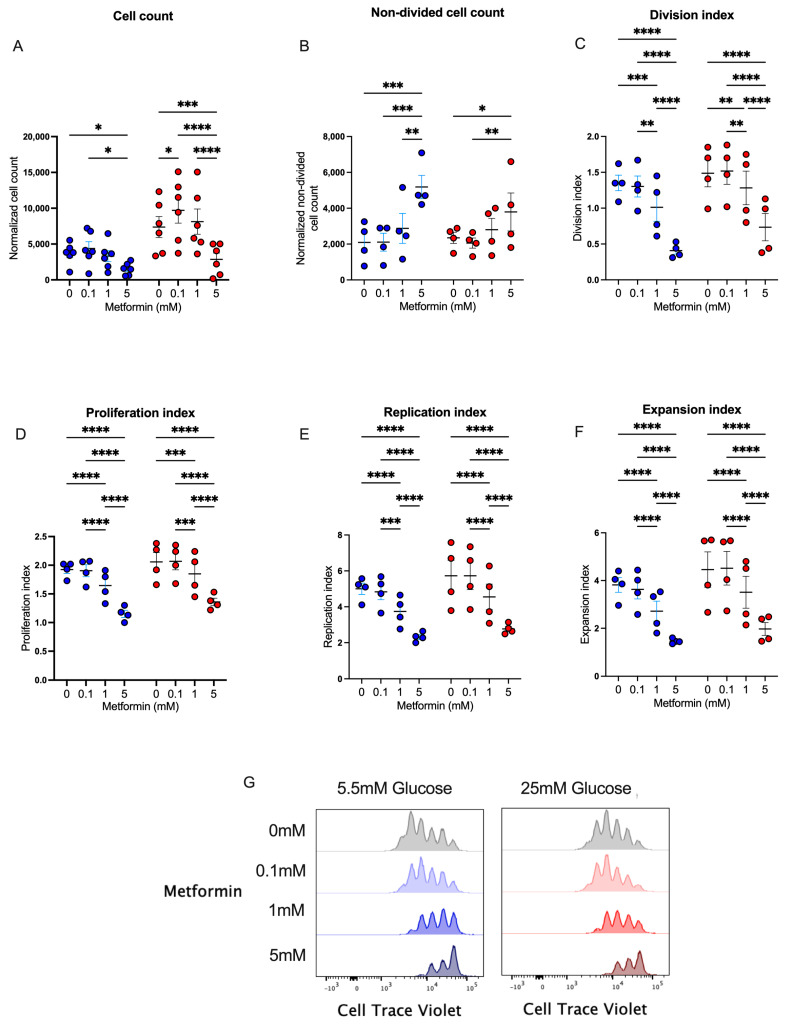
Metformin affects cell counts and proliferation of in vitro-activated CD4+ T lymphocytes under normal and high glucose. (**A**) Cell count and (**B**) non-divided cell count of CD4+ T lymphocytes under glucose (5.5 mM and 25 mM) and metformin (0 mM, 0.1 mM, 1 mM, 5 mM) conditions. (**C**) Division index, (**D**) proliferation index, (**E**) replication, (**F**) expansion index, and (**G**) histograms of Cell Trave Violet™ of CD4+ T lymphocytes under glucose (5.5 mM and 25 mM) and metformin (0 mM, 0.1 mM, 1 mM, 5 mM) conditions. (**A**–**F**) Blue points represent glucose 5.5 mM and red points 25 mM. * (*p* < 0.05), ** (*p* < 0.01), *** (*p* < 0.001), and **** (*p* < 0.0001).

**Figure 2 biomolecules-14-00846-f002:**
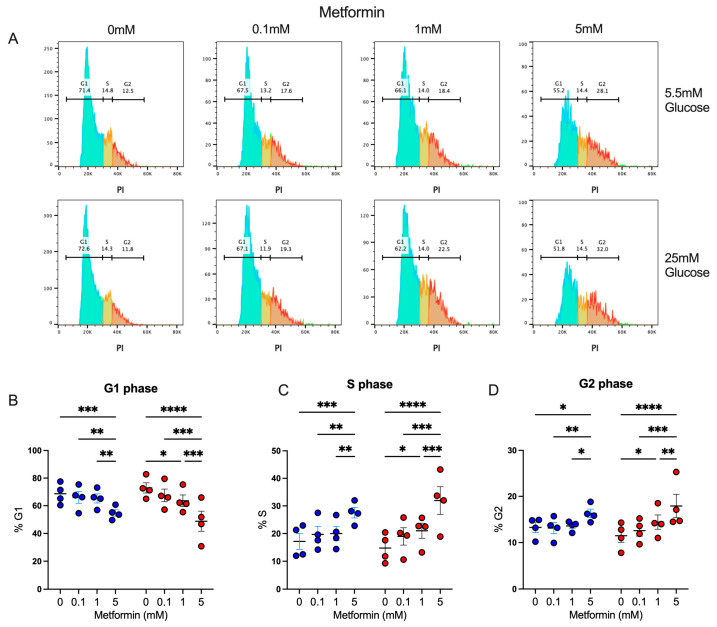
Metformin arrest cell cycle of in vitro activated CD4+ T lymphocytes under normal and high glucose. (**A**) Propidium iodide histograms, and percentage of cells in (**B**) G1 phase, (**C**) S phase, and (**D**) G2 phase of CD4+ T lymphocytes under glucose (5.5 mM and 25 mM) and metformin (0 mM, 0.1 mM, 1 mM, 5 mM) conditions. (**B**–**D**) Blue points represent glucose 5.5 mM and red points 25 mM. * (*p* < 0.05), ** (*p* < 0.01), *** (*p* < 0.001), and **** (*p* < 0.0001).

**Figure 3 biomolecules-14-00846-f003:**
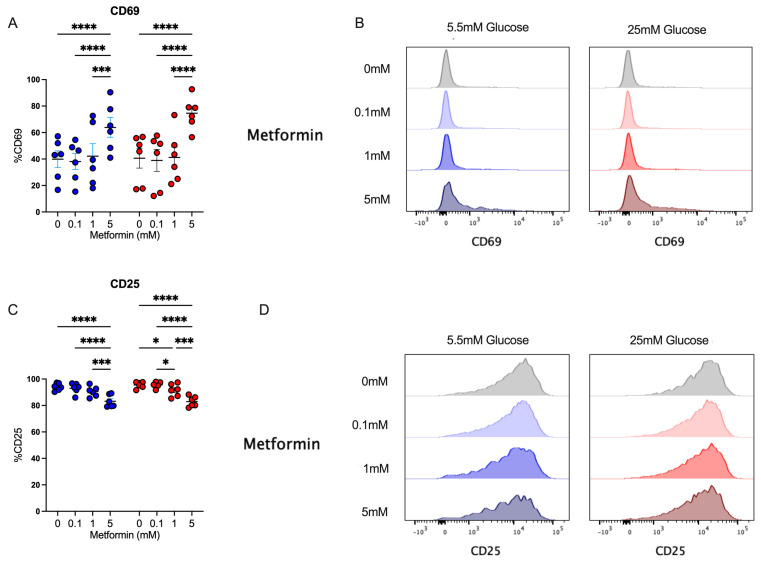
Despite affecting cell cycle, metformin up-regulates the activation marker CD69 in CD4+ T lymphocytes in vitro-activated under normal and high glucose. (**A**,**B**) CD69+ and (**C**,**D**) CD25+ of CD4+ T lymphocytes under glucose (5.5 mM and 25 mM) and metformin (0 mM, 0.1 mM, 1 mM, 5 mM) conditions. (**A**,**C**) Blue points represent glucose 5.5 mM and red points 25 mM. * (*p* < 0.05), *** (*p* < 0.001), and **** (*p* < 0.0001).

**Figure 4 biomolecules-14-00846-f004:**
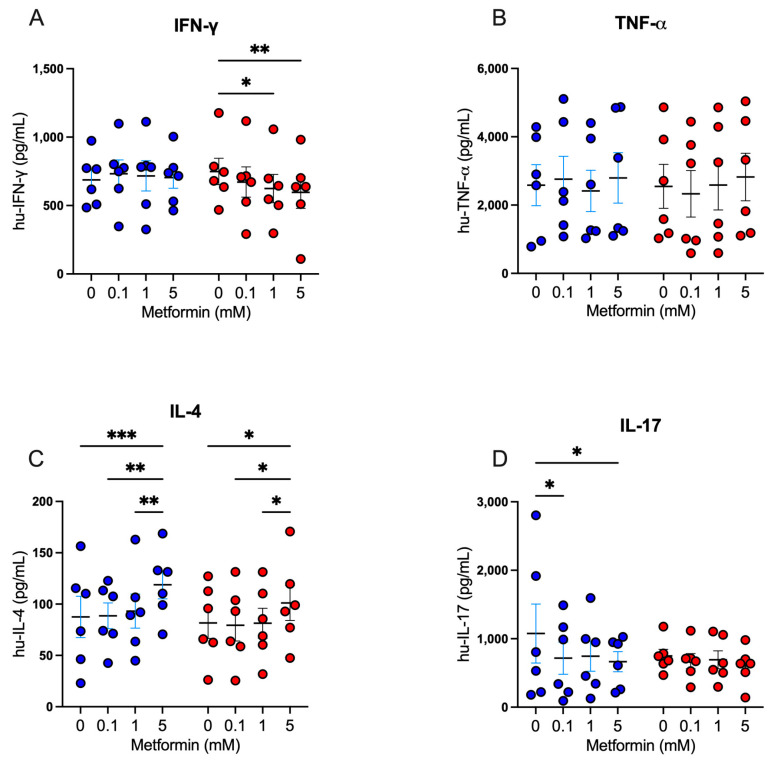
Metformin modulates cytokine secretion in activated CD4+ T lymphocytes. (**A**) IFN-g, (**B**) TNF-a, (**C**) IL-4, and (**D**) IL-17 secretion of CD4+ T lymphocytes under glucose (5.5 mM and 25 mM) and metformin (0 mM, 0.1 mM, 1 mM, 5 mM) conditions. (**A**–**D**) Blue points represent glucose 5.5 mM and red points 25 mM. * (*p* < 0.05), ** (*p* < 0.01), and *** (*p* < 0.001).

**Figure 5 biomolecules-14-00846-f005:**
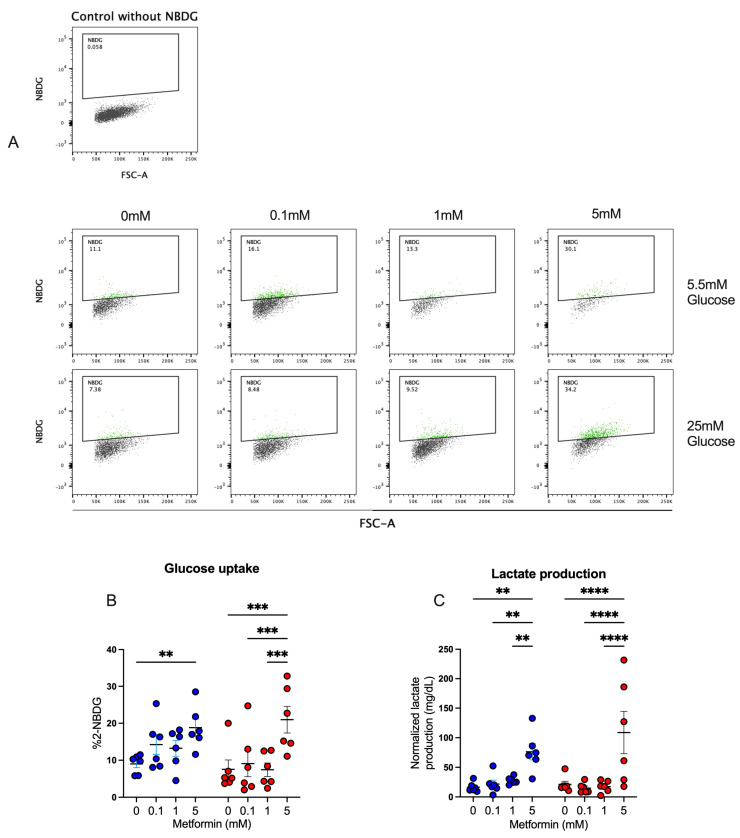
Metformin promotes glucose uptake and lactate production in activated CD4+ T lymphocytes. (**A**,**B**) Glucose uptake and (**C**) lactate production of CD4+ T lymphocytes under glucose (5.5 mM and 25 mM) and metformin (0 mM, 0.1 mM, 1 mM, 5 mM) conditions. (**B**,**C**) Blue points represent glucose 5.5 mM and red points 25 mM. ** (*p* < 0.01), *** (*p* < 0.001), and **** (*p* < 0.0001).

## Data Availability

The data presented in this study are available on request from the corresponding author due to ethical reasons.

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
