# Peer review of "Metformin Restrains the Proliferation of CD4+ T Lymphocytes by Inducing Cell Cycle Arrest in Normo- and Hyperglycemic Conditions"

_biomolecules, 2024, doi:10.3390/biom14070846_

Round 1

Reviewer 1 Report

Comments and Suggestions for Authors

  In this manuscript, Cartes-Velásquez R. et al. investigated the effect of metformin in CD4+ T lymphocytes. They found that metformin inhibits cell proliferation without affecting cell activation, cytokine production and glucose metabolism. Data is interesting, but several points need to be improved for publication.

1 How many healthy controls did they isolated CD4+ T lymphocyte ? Describe in Materials and Methods (2.1)

2 What is the meaning of red and blue circle in Figure 1-5 ? Probably I think red means 5.5 mM glucose and blue circle means 25 mM glucose, but they did not mention. Describe them in Figure legends.

3 Unify the format of IFN-g (Check line 72, 241, 253, 359 and 364) and TNF-a( Check line 242 and 253).

4  In references, the numbers reference are duplicated. Correct them. 

Comments on the Quality of English Language

Although some points need to be corrected, English quality is good.

Author Response

Dear Editor,

We would like to express our gratitude to both reviewers for their valuable comments and the opportunity to enhance our manuscript. In this revised version, we have addressed all the corrections suggested by the reviewers and have conducted the intracellular cytokine staining coupled with flow cytometry as requested by reviewer 2.

Regards,

Reviewer 1

In this manuscript, Cartes-Velásquez R. et al. investigated the effect of metformin in CD4+ T lymphocytes. They found that metformin inhibits cell proliferation without affecting cell activation, cytokine production and glucose metabolism. Data is interesting, but several points need to be improved for publication.

1 How many healthy controls did they isolated CD4+ T lymphocyte ? Describe in Materials and Methods (2.1)

R: We isolated CD4+ T lymphocytes from six healthy patients who did not have any diseases. This information has been included in the Materials and Methods section (2.1).

2 What is the meaning of red and blue circle in Figure 1-5 ? Probably I think red means 5.5 mM glucose and blue circle means 25 mM glucose, but they did not mention. Describe them in Figure legends.

R: The red circle represents 5.5 mM glucose, while the blue circle represents 25 mM glucose. We have clarified this in the figure legends.

3 Unify the format of IFN-g (Check line 72, 241, 253, 359 and 364) and TNF-a( Check line 242 and 253).

R: The format of IFN-g and TNF-a has been unified across the entire paper as per your suggestion.

4  In references, the numbers reference are duplicated. Correct them. 

R: We have reviewed the references and corrected all instances of duplicated reference numbers.

Reviewer 2 Report

Comments and Suggestions for Authors

General Comments: The study by Cartes-Velásquez et al. exploring the impact of Metformin on CD4+ T cell proliferation and effector functions provides valuable insights into the immunomodulatory potential of a widely used antidiabetic drug. The implications of such findings are significant, offering a potential link between metabolic interventions and immune regulation.  

Specific Comments:  

1.     The inclusion of Tregs within the isolated CD4+ T cell population is a pertinent consideration, given their critical role in maintaining immune homeostasis. Previous research has indeed investigated the effects of Metformin on Treg cells as also highlighted by the authors. It would be beneficial for the authors to clarify whether Tregs were present in their isolated populations and, if so, to discuss the potential influence of Metformin on these cells in the context of their findings.  

2.     The concern regarding the reduced cytokine production being a consequence of diminished CD4+ T cell proliferation rather than a direct effect on cytokine synthesis is valid. To address this, additional analyses such as real-time PCR for gene expression or intracellular cytokine staining coupled with flow cytometry would provide more definitive insights into the individual cell-level cytokine production.

3.     The data showed that Metformin induced similar regulation to CD4+ T cells under low and high glucose level. The author should discuss the clinical implication about this.

4.     The authors' initial mention that Metformin generally inhibits CD4+ T cell proliferation due to the reduction of the glycolytic pathway appears to contrast with their subsequent data showing an increase in glucose uptake and lactate production in CD4+ T cells following Metformin treatment. This discrepancy merits a thorough discussion. Clarifying these points would enhance the understanding of Metformin's complex role in immune cell metabolism and function.

Author Response

Dear Editor,

We would like to express our gratitude to both reviewers for their valuable comments and the opportunity to enhance our manuscript. In this revised version, we have addressed all the corrections suggested by the reviewers and have conducted the intracellular cytokine staining coupled with flow cytometry as requested by reviewer 2.

Regards,

Reviewer 2

General Comments: The study by Cartes-Velásquez et al. exploring the impact of Metformin on CD4+ T cell proliferation and effector functions provides valuable insights into the immunomodulatory potential of a widely used antidiabetic drug. The implications of such findings are significant, offering a potential link between metabolic interventions and immune regulation.  

R: Thank you for the comment.

Specific Comments:  

1.     The inclusion of Tregs within the isolated CD4+ T cell population is a pertinent consideration, given their critical role in maintaining immune homeostasis. Previous research has indeed investigated the effects of Metformin on Treg cells as also highlighted by the authors. It would be beneficial for the authors to clarify whether Tregs were present in their isolated populations and, if so, to discuss the potential influence of Metformin on these cells in the context of their findings.  

R: We isolated CD4+ T cells using magnetic beads and further purified them with cell sorting, thus including the Treg subpopulation in our samples. Although we did not assess specific subpopulations, we have added a paragraph in the Discussion based on literature review. Generally, we anticipated a relative increase in Tregs with metformin use, possibly linked to a reduction in effector T cells.

2.     The concern regarding the reduced cytokine production being a consequence of diminished CD4+ T cell proliferation rather than a direct effect on cytokine synthesis is valid. To address this, additional analyses such as real-time PCR for gene expression or intracellular cytokine staining coupled with flow cytometry would provide more definitive insights into the individual cell-level cytokine production.

R: We appreciate the suggestion. An assessment of intracellular cytokine staining coupled with flow cytometry was conducted, encompassing two metformin concentrations (0mM and 4mM) and two glucose concentrations (5.5mM and 25mM). The plots from this assay have been uploaded as a supplemental file. Notably, three healthy donors, distinct from those in the original experiments, were included.

Regarding IL-4, we observed similarities in levels between the media and intracellular compartments. However, for IFN-α, while lower intracellular levels were noted with metformin at 5.5mM glucose, two out of three donors exhibited higher levels at 25mM glucose, contradicting the levels measured in the media. For IL-17, we encountered contradictory results, with metformin inducing higher intracellular levels but lower levels in the media.

Interpreting these results is intricate and goes beyond the scope of this study. Nevertheless, aligning with the reviewer's suggestion, we hypothesize that cytokine levels could be influenced by factors such as cell proliferation and/or cytokine release into the extracellular space rather than cytokine production.

3.     The data showed that Metformin induced similar regulation to CD4+ T cells under low and high glucose level. The author should discuss the clinical implication about this.

R: Thank you for raising this point. We have enhanced our analysis of this matter in the Discussion section. We hypothesize that the short exposure period (4 days) and the absence of immunological challenges (infection, cancer cells, etc.) need consideration in extrapolating these results to diabetes or other hyperglycemic conditions. 

4.     The authors' initial mention that Metformin generally inhibits CD4+ T cell proliferation due to the reduction of the glycolytic pathway appears to contrast with their subsequent data showing an increase in glucose uptake and lactate production in CD4+ T cells following Metformin treatment. This discrepancy merits a thorough discussion. Clarifying these points would enhance the understanding of Metformin's complex role in immune cell metabolism and function.

R: We acknowledge the discrepancy and have revised the Discussion to address it. Previous evidence indeed presents contradictory findings regarding the impact of metformin on the glycolytic pathway in CD4+ T cells. Some studies have reported no reduction or even an increase in glycolysis following metformin exposure, which is now included in the Discussion.

Round 2

Reviewer 2 Report

Comments and Suggestions for Authors

The comments were adequately addressed.